# Quantum Advantage of Thermal Machines with Bose and Fermi Gases

**DOI:** 10.3390/e25020372

**Published:** 2023-02-17

**Authors:** Saikat Sur, Arnab Ghosh

**Affiliations:** 1Department of Chemical and Biological Physics, Weizmann Institute of Science, Rehovot 7610001, Israel; 2Department of Chemistry, Indian Institute of Technology, Kanpur 208016, India

**Keywords:** quantum thermodynamics, heat engine, refrigerator, Fermi gas, Bose gas

## Abstract

In this article, we show that a quantum gas, a collection of massive, non-interacting, indistinguishable quantum particles, can be realized as a thermodynamic machine as an artifact of energy quantization and, hence, bears no classical analog. Such a thermodynamic machine depends on the statistics of the particles, the chemical potential, and the spatial dimension of the system. Our detailed analysis demonstrates the fundamental features of quantum Stirling cycles, from the viewpoint of particle statistics and system dimensions, that helps us to realize desired quantum heat engines and refrigerators by exploiting the role of quantum statistical mechanics. In particular, a clear distinction between the behavior of a Fermi gas and a Bose gas is observed in one dimension, rather than in higher dimensions, solely due to the innate differences in their particle statistics indicating the conspicuous role of a quantum thermodynamic signature in lower dimensions.

## 1. Introduction

The study of quantum thermodynamics comprises the basis for analyzing heat engines and refrigerators at the microscopic level [1,2,3,4,5]. Although classical thermodynamics is extremely successful in predicting the statistical behavior of a classical gas, a fundamental departure from classical mechanics is absolutely necessary to fully understand the thermodynamic behavior of multi-particle quantum thermal machines [6,7,8,9,10,11]. In fact, one might expect new thermodynamic effects would emerge in this case without having any classical correspondence. In particular, it is quite expected that many body quantum thermal machines must be reconciled in view of energy quantization, degeneracies, and, most importantly, the interplay between particle statistics and interaction potential [6,7,8,9,12,13,14,15,16].

In recent years, several proposals for constructing quantum mechanical versions of thermal machines, such as Otto, Carnot, Stirling, and Diesel [17,18,19,20,21,22,23,24,25,26,27,28,29,30,31], with different working media, like a particle in a box, harmonic oscillator and spin systems [12,20,32,33,34,35,36,37,38,39,40,41,42,43], have been presented. Most of the cases that have been analyzed for understanding the generic thermodynamic features in quantum thermal machines do not exhibit any proper evidence of quantum advantage over their classical counterparts [44,45,46,47,48]. Among them, only a few analyze the quantum-enhanced machines specifically from the viewpoint of a single particle within a potential well, based on quantized energy levels [12,13,49,50]. On the other hand, many-particle quantum thermal machines were shown to outperform the classical counterparts using interacting Bose gases in a time-dependent harmonic trap and a tight waveguide [7,9]. In these studies, the volume equivalent of the heat machines is taken to be the frequency of the harmonic oscillator and the inter-particle interaction strength, respectively. Both these studies conveyed the quantum supremacy arising from the interplay between many-particle quantum effects. The role of quantum statistics comes into play if the system comprises an ensemble of identical particles in contact with a pair of low-temperature baths. In this case, the system dynamics is governed by the principle of how a single particle state is occupied. This distinguishes the behavior of two different types of particles, viz. fermions, and bosons. The chemical potential that depends on the dimensionality of the system, in addition to its statistics, can play a huge role in determining its behavior. Therefore, the collective behavior of quantum particles is interlinked with the dimensionality and, hence, the degeneracy of the energy levels [51,52,53].

In this article, we elaborately, rigorously, and methodically study the dependence of particle statistics and dimensionality of the system in the context of a quantum Stirling cycle, based on an infinite potential box. In the present scenario, when the potential is distorted by introducing an infinite barrier in the middle of the box, the odd levels shift upwards and overlap with the even energy levels that are immediately next, creating a new energy level structure with degeneracies (see e.g., Figure 1b) [54,55]. In quantum Stirling cycles, this distortion of the potential may constitute a certain amount of work and heat exchange between the baths, exclusively based on quantized energy levels, without having any classical analog of volume [56,57]. Depending on the behavior of the cycle, whether work is extracted or not, one can construct a desired Stirling heat engine or refrigerator just by exploiting the quantum nature of the identical particles. The approach to a many-particle quantum system with symmetrizing/anti-symmetrizing wavefunctions is ineffective, especially in the N→∞ limit, where distinct statistical behavior is observed. Here, we show that one can bypass the problem by using the tools of quantum statistical mechanics.

Our approach is interesting, both from theoretical and practical viewpoints, as recent advances in realizing heat machines with quantum components pave the way for the realization of multi-particle energy quantization-assisted machines in future [26,37,45,58,59,60,61,62]. Particularly, quantum dots (electrons or holes confined in three spatial dimensions), wires (electrons or holes confined in two spatial dimensions and free in the other direction), and wells (electrons or holes confined in one spatial dimension but free in the other two directions) [63,64,65,66] and ion-trap systems [67,68,69], are potential candidates for realizing quantum particles entrapped in a potential well.

This paper is organized as follows: In Section 2, we present the basic formulation, with a brief overview of the quantum Stirling cycle, followed by detailed characteristics of the working medium and the fundamental principles behind quantum statistics of identical particles. In Section 3, we summarize the main results and their interpretations, together with the analytical approach, in the limiting case of a large number of particles and extremely low temperature. Finally, we conclude in Section 4.

## 2. Formulation

### 2.1. Quantum Stirling Cycle

A quantum Stirling cycle [13] consists of two configurations, governed by two different Hamiltonians, *H* and $H^{\prime }$, respectively, and each configuration is kept in equilibrium with two thermal baths. Hence, the cycle has four stages, each connected to its preceding and succeeding stages through two isothermal and two isochoric processes, as shown schematically in Figure 1a. Initially, at stage *A*, the system is in equilibrium with a hot bath at a temperature $T_{h}$. In the first step (A→B), the potential is distorted quasi-statically and isothermally to change the configurations of the energy levels and the wavefunctions, while the system is still coupled to the hot bath (Figure 1b). In the next step (B→C), the system is detached from the hot bath and connected with a cold bath at a temperature $T_{c}$, via an isochoric process. Consequently, the system fully thermalizes with the cold bath, and the temperature of the system falls from $T_{h}$ to $T_{c}$, without performing any mechanical work. In the next step, (C→D), the system is brought back to its original configuration quasi-statically and isothermally, while keeping the system connected to the cold bath. In the final step (D→A), the system is detached from the cold bath and connected to the hot bath through an isochoric process, without performing any mechanical work. The system again fully thermalizes with the hot bath, and the temperature rises from $T_{c}$ to $T_{h}$.

The total heat transferred to the system from the bath in the isothermal processes (A→B) and (C→D), while keeping the system in equilibrium with the bath at temperatures $T_{h}$ and $T_{c}$, respectively, are combinations of the mechanical work performed due to deformation of the potential and the change of the internal energies. Thus, the heat transferred to the bath from the system during two isothermal processes is given by:(1)$$\begin{array}{c} Q_{AB}=-\left(U_{B}-U_{A}+k_{B}T_{h}\ln Z_{B}-k_{B}T_{h}\ln Z_{A}\right), \end{array}$$
and
(2)$$\begin{array}{c} Q_{CD}=-\left(U_{D}-U_{C}+k_{B}T_{c}\ln Z_{D}-k_{B}T_{c}\ln Z_{C}\right), \end{array}$$
respectively. By definition, the internal energy at a temperature *T*, in terms of the thermal partition function $Z_{T}$, is given as:(3)$$U_{T}=k_{B}T^{2}\frac{\partial }{\partial T}\ln Z_{T}.$$
In what follows in Section 2.3, we explicitly evaluate the thermal partition functions of the Stirling cycle with a non-interacting quantum gas of fermions and bosons. On the contrary, the heat transferred to the system during the isochoric processes, (B→C) and (D→A), are only the differences between the average energies of the initial and the final configurations:(4)$$\begin{array}{c} Q_{BC}=-\left(U_{C}-U_{B}\right),\;\;\mathrm{and}\;\;Q_{DA}=-\left(U_{A}-U_{D}\right). \end{array}$$

Provided all the processes involved in the cycle are reversible, and no leakage takes place, the expressions for the net work done on the system *W* and the heat transferred to the system with the hot and cold baths, $Q_{h}$ and $Q_{c}$, after completion of one cycle are given as follow:(5)$$\begin{array}{ccc} W & = & \left(Q_{AB}+Q_{BC}+Q_{CD}+Q_{DA}\right)=-k_{B}\left(T_{h}\ln\frac{Z_{B}}{Z_{A}}-T_{c}\ln\frac{Z_{C}}{Z_{D}}\right),\\ Q_{h} & = & -\left(Q_{AB}+Q_{DA}\right)=k_{B}T_{h}\ln\frac{Z_{B}}{Z_{A}}+U_{B}-U_{D},\\ Q_{c} & = & -\left(Q_{BC}+Q_{CD}\right)=-k_{B}T_{c}\ln\frac{Z_{C}}{Z_{D}}-U_{B}+U_{D}. \end{array}$$
By the principle of conservation of energy, or the first law of thermodynamics, one can check that $Q_{h}+Q_{c}+W=0$. In our convention, if the quantities, $Q_{h}$ or $Q_{c}$, are positive, the heat flows into the system; similarly, if work, *W*, is positive, work is done on the system. In conformity with the second law of thermodynamics, only four modes of operation are possible. The four possible modes can be identified by the signs of *W*, $Q_{h}$, and $Q_{c}$, as depicted in Table 1.

Now, if heat is absorbed from the hot bath at a higher temperature by the system, it converts a fraction of it into work, and rejects the rest to the bath at a lower temperature, we term this a heat engine mode. The system works as a heat engine if *W* is negative, i.e., work can be extracted from the system. The efficiency $\eta_{E}$ of the engine, defined as the ratio of the work extracted to the total heat absorbed by the system, is:(6)$$\begin{array}{c} \;\eta_{E}=-\frac{W}{Q_{h}}. \end{array}$$

In contrast, if heat is absorbed from the cold bath and rejected into the hot bath with external work done on the system, we term it refrigerator mode. Here, *W* is positive, i.e., work is done on the system. The coefficient of performance of a refrigerator, the ratio of the heat extracted from the cold bath to work done on the system, is given as:(7)$$\begin{array}{c} \eta_{R}=\frac{Q_{c}}{W}. \end{array}$$

Apart from the two modes discussed, there can be two other modes. In the accelerator mode, there is a free flow of heat from the hot bath to the cold bath, and work is done on the system. On the other hand, in the heater mode, a fraction of the work done on the system is rejected to the hot bath and the rest to the cold bath.

The Carnot cycle, a theoretical proposition of Carnot, consists of only reversible processes and, therefore, conserves the total entropy, while the system completes one cycle. However, it provides the maximum possible efficiency for a given pair of thermal baths, which is solely dependent on the bath temperatures [70]. In order to provide an estimate of its performance, we present the efficiency and coefficient of performance of the quantum cycle scaled by the same to its Carnot equivalent. The efficiency of a Carnot engine and the coefficient of performance of the Carnot refrigerator are given, respectively as:(8)$$\begin{array}{c} \eta_{E}^{max}=1-\frac{T_{c}}{T_{h}},\;\;and\;\;\eta_{R}^{max}=\frac{T_{c}}{T_{h}-T_{c}}. \end{array}$$

The work done during a cycle primarily depends on the potential deformation that changes the wavefunction of the working substance and shifts the energy eigenvalues along with the bath temperatures. As discussed, quasi-static deformation of the potential can cause extraction of work from the system and, hence, serve as a quantum heat engine or a refrigerator, depending on whether heat is transferred from a hot bath to a cold bath, and vice versa. As a practical example of the aforementioned situations, we considered a quantum thermal machine that uses a gas of quantum particles in a hard potential box as its working medium [12,13], and discuss the details of its properties.

Although in this article, we demonstrate our results for a Stirling cycle, one can study other thermodynamic cycles with Bose and Fermi gases, case by case. For example, an Otto cycle consists of a pair of isochoric and isentropic (adiabatic) processes. In our context, the barriers were to be introduced adiabatically and the rest of the steps were identical. With the approximation of adiabaticity, which ensured that the level populations did not change upon removal of the bath, one could have results similar to that of a Stirling engine [12]. One could generalize the results by replacing a single particle working medium with a quantum gas with fermions and bosons. Whereas an exact equivalence with a Carnot cycle was difficult to construct in this setup, since there should be a pair of expansion steps, an isothermal and an adiabatic.

### 2.2. The Working Medium

We considered an ensemble of non-interacting, massive, indistinguishable particles of mass *M* confined in a *d* dimensional potential box with length 2L in each dimension as our working medium of the thermal machine. As the particles were non-interacting, the full Hamiltonian of the system was the sum of identical single-particle Hamiltonians. The single particle Hamiltonian is given as:(9)$$H=\sum_{i=1}^{d}\frac{p_{x_{i}}^{2}}{2M},\;\;\;for\;\left|x_{i}\right|<L.$$
The potential was distorted by introducing *d* impenetrable barriers at $x_{i}=0$, one in each dimension, that changed the energy levels and distorted the wavefunctions. In this context, to highlight the quantum advantage over its classical counterpart, we noted that introducing delta function barriers did not change the volume of the classical system and, thereby, had zero contribution to the work. In addition to that, if the box became very large, gaps between the energy levels went zero, forming free particles with a continuum of energy levels.

The single-particle Hamiltonian of the system in the distorted configuration, with *d* impenetrable barriers, is given by:(10)$$H^{\prime }=\sum_{i=1}^{d}\frac{p_{x_{i}}^{2}}{2M}+\lambda \delta \left(x_{i}\right),\;\;for\;\left|x_{i}\right|<L,$$
where λ is the strength of the delta function barrier. The single particle eigen energies of the Hamiltonians, *H* and $H^{\prime }$, labeled by a set of integer quantum numbers $n_{x_{i}}$, are given by the following expressions: (11)$$\begin{array}{ccc} E(n_{x_{1}},n_{x_{2}},\ldots ,n_{x_{d}}) & = & \frac{\pi^{2}\hbar^{2}}{8ML^{2}}\sum_{i=1}^{d}n_{x_{i}}^{2};\;n_{x_{i}}=1,2,3,\ldots \\ E^{\prime }\left(n_{x_{1}},n_{x_{2}},\ldots ,n_{x_{d}}\right) & = & \frac{\pi^{2}\hbar^{2}}{8ML^{2}}\sum_{i=1}^{d}{\left[n_{x_{i}}+\frac{\varepsilon (n_{x_{i}})}{2}\left(1-{\left(-1\right)}^{n_{x_{i}}}\right)\right]}^{2};\;n_{x_{i}}=1,2,3,\ldots \end{array}$$
where $0\leq \varepsilon (n_{x_{i}})\leq 1$. The value of $\varepsilon (n_{x_{i}})$ depends on the barrier strength λ and can be obtained from the graphical solution of the transcendental equation (see [54,55] for details). Since we considered, here, an infinitely strong barrier, i.e., λ→∞, the quantity $\varepsilon (n_{x_{i}})=1$. Inserting an impenetrable barrier shifted each odd energy eigenstate towards its next even one and, thus, created twice degenerate energy levels. The single-particle energy levels and eigenfunctions are schematically shown in Figure 1b. In a system with *d* dimensions, one can show that there are $2^{d}$ degenerate energy levels [54]. For example, in two dimensions, the energy levels given by Equation (Equation 11) are denoted by a pair of quantum numbers $\left(n_{x_{1}},n_{x_{2}}\right)$. Now, without the barriers, the pair $\left(n_{x_{1}},n_{x_{2}}\right)$ take values (1,1),(1,2),(2,1),(2,2),(1,3),(2,3),(1,4),(2,4),… and so on. After introducing two impenetrable barriers at $x_{1}=0$ and $x_{2}=0$, the energies of the same set change to the energy levels corresponding to the quantum numbers (2,2),(2,2),(2,2),(2,2),(2,4),(2,4),(2,4),(2,4),… of the undeformed box and, thus, introduce a degeneracy of 4.

The approach of dealing with a large number of quantum particles in the system by symmetrizing/anti-symmetrizing their states is inefficient and intractable, even in computers, particularly in the limit of an average number of particles, N→∞. However, the collective effects of bosons or fermions are expected to be prominent in the said limit. At this juncture, the problem can be bypassed using the tools of quantum statistical mechanics.

A critical viewpoint in this context is worth mentioning. It has been shown that effective inter-particle interactions play an important role in determining the qualitative thermodynamic behavior of a many-particle system in one dimension [7]. A gas of hard-core (strongly repulsive) bosons, a gas of ideal fermions, and a gas of hardcore anyons in one dimension behave identically owing to their Bose–Fermi and anyon–anyon dualities [71,72]. Once the restriction of the hard-core condition is relaxed, one might anticipate a thermodynamic signature of bosons and fermions to emerge, as we demonstrate in this paper. For higher dimensions, the compressibility of the working medium plays the same role as inter-particle interaction does in one dimension. For details, refer to the very interesting articles by Şişman et al. [6] and Jaramillo et al. [7].

### 2.3. Statistics of Quantum Particles

Quantum statistical mechanics is the foundation of understanding the low-temperature behavior of multi-particle physical systems consisting of identical particles. In quantum mechanics, all identical particles are classified into two categories; the class of particles with half-integer spin, known as fermions, is described by Fermi–Dirac statistics, and the other class, with integer spins, known as bosons, is described by Bose–Einstein statistics. The fundamental difference between these two categories arises entirely from the principle of how a single particle state is occupied. Two fermions can never occupy a single particle state; on the other hand, multiple bosons can occupy the same state.

Quantum statistics dominate only when the inter-particle spacing becomes smaller than the thermal de Broglie wavelength under low temperature, which can be termed “*quantum regime*”. On the contrary, both statistics can be approximated by Maxwell–Boltzmann distribution in the classical regime, where quantum effects are negligible. The thermal grand partition functions $Z_{T}^{+}$ and $Z_{T}^{-}$ for *N* non-interacting fermions and bosons at a temperature *T* are, respectively [73,74]:(12)$$\begin{array}{ccc} Z_{T}^{+} & = & \prod_{n=1}^{\infty }{\left(\sum_{j=0}^{1}e^{-j({\tilde{E}}_{n}-\tilde{\mu })/T}\right)}^{g_{n}}=\prod_{n=1}^{\infty }{\left(1+e^{-({\tilde{E}}_{n}-\tilde{\mu })/T}\right)}^{g_{n}},\\ Z_{T}^{-} & = & \prod_{n=1}^{\infty }{\left(\sum_{j=0}^{\infty }e^{-j({\tilde{E}}_{n}-\tilde{\mu })/T}\right)}^{g_{n}}=\prod_{n=1}^{\infty }{\left(\frac{1}{1-e^{-({\tilde{E}}_{n}-\tilde{\mu })/T}}\right)}^{g_{n}}, \end{array}$$
where $g_{n}$ is the degeneracy of the single particle *n*-th energy level and $\tilde{\mu }$ is the scaled chemical potential. Note that, in the above equation, the dependence on the particle number is hidden in the chemical potential. For bosons, it is to be noted that the *N* dependence on $Z_{T}^{-}$ vanishes as the constraint on the particle number is relaxed in the grand canonical ensemble. For the sake of simplicity, we express all the energies and the chemical potentials in the units of $k_{B}$, i.e., $\tilde{E}=E/k_{B}$ and $\tilde{\mu }=\mu /k_{B}$ in Equation (Equation 12).

As is seen in the next section, the chemical potential of the system plays an essential role in determining the properties of the thermal machine. Now, to estimate the value of the chemical potential for fermions, in terms of an average number of particles *N*, it is important to define the Fermi energy $E_{F}$, the energy of the highest filled state at T=0, which is given as $E_{F}=\frac{\hbar^{2}k_{F}^{2}}{2M}$ where ${\vec{k}}_{F}$ is the Fermi wave vector. In *d*-dimension, the relation [51,52,53,75]$\frac{C_{d}k_{F}^{d}}{{\left(2\pi /2L\right)}^{d}}=N$ yields $k_{F}=2\pi {\left(\frac{\sigma }{C_{d}}\right)}^{1/d}$. Hence the dimensionless Fermi energy, in terms of the volume of a *d*-dimensional unit sphere [53,76], $C_{d}=\frac{\pi^{d/2}}{\Gamma (d/2+1)}$ and the particle density $\sigma =\frac{N}{{\left(2L\right)}^{d}}$, can be expressed as:(13)$$\begin{array}{c} {\tilde{E}}_{F}=\frac{E_{F}}{k_{B}}=2\alpha L^{2}{\left(\frac{\sigma }{C_{d}}\right)}^{2/d}=\frac{\alpha N^{2/d}}{2C_{d}^{2/d}}. \end{array}$$

For example,
(14)$$\begin{array}{ccc} & & {\tilde{E}}_{F}^{\left(d=1\right)}=\frac{\alpha L^{2}\sigma^{2}}{2}=\frac{\alpha N^{2}}{8},\\ & & {\tilde{E}}_{F}^{\left(d=2\right)}=\frac{2\alpha L^{2}\sigma }{\pi }=\frac{\alpha N}{2\pi },\\ & & {\tilde{E}}_{F}^{\left(d=3\right)}={\left(\frac{6}{\pi }\right)}^{2/3}\frac{\alpha L^{2}\sigma^{2/3}}{2}={\left(\frac{6}{\pi }\right)}^{2/3}\frac{\alpha N^{2/3}}{8}. \end{array}$$
At extremely low temperatures, for fermions, all the energy levels below the Fermi energy are occupied. Introducing barrier(s) in the system does not change the Fermi energy appreciably. We, therefore, can consider the chemical potentials in the two configurations, with and without the barrier(s), to be equal, i.e., $\mu =\mu^{\prime }$.

For the convenience of our calculations, here, we defined a parameter $\alpha =\frac{\hbar^{2}\pi^{2}}{k_{B}ML^{2}}$ with the dimension of temperature ($\left[\alpha \right]={\left[\hbar \right]}^{2}{\left[k_{B}\right]}^{-1}M^{-1}L^{-2}={\left(ML^{2}T^{-1}\right)}^{2}{\left(ML^{2}T^{-2}\Theta^{-1}\right)}^{-1}M^{-1}$
$L^{-2}=\Theta$) such that the low-temperature condition was scaled w.r.t α as $T_{h,c}/\alpha ≲1$. We re-emphasize here that, as the bath temperatures $T_{h},T_{c}$ became greater than α, the energy levels became continuous, and the quantum advantage was gradually lost; we term this limit “classical regime”. In order to get an estimate of the different parameters related to the system, let us consider a system of electrons entrapped in a potential well. In terms of the fundamental constants, α=1 corresponds to a box length L∼100 nm. Since α is the characteristic temperature associated with the system, the low-temperature limit, in this case, reduces to $T_{h},T_{c}≲1K$. From the expression of α, it is evident with the decrease of the box size, i.e., the more confined the quantum system becomes, the constraint on the two bath temperatures becomes more relaxed.

Finally, we note that, in the low-temperature limit, the chemical potential for fermions can be taken to be equal to the Fermi energy, as the difference is extremely small, even at room temperature [53,77].

In the case of bosons, one can easily observe, from the Bose–Einstein distribution, that the chemical potential must be smaller than the ground state of the system in order to ensure a positive value for the occupation number. As the ground state energies of a system with *d* dimensions are $E_{1}=\alpha d/8$ and $E_{1}^{\prime }=\alpha d/2$ for the configurations without, and with, the barriers, respectively, the corresponding chemical potentials must satisfy the conditions μ≤αd/8 and $\mu^{\prime }\leq \alpha d/2$. Hence, the relation between the chemical potential and the average particle number becomes involved in this case. Starting from the boundary condition that the sum over the occupation numbers $N_{n}$ of all energy levels equals the total number of particles, i.e., $N=\sum_{n}N_{n}$, one can write:(15)$$\begin{array}{ccc} N & & =\sum_{n=1}^{\infty }\frac{g_{n}}{e^{({\tilde{E}}_{n}-\tilde{\mu })/T}-1}. \end{array}$$

In the extremely low-temperature limit, all the terms, except the one corresponding to the ground state, are negligible. With this approximation, the *N* dependence of the chemical potential for the bosons is given by the following form:(16)$$\begin{array}{c} \tilde{\mu }\approx {\tilde{E}}_{1}-T\ln\left(1+\frac{g_{1}}{N}\right). \end{array}$$

For a system of *d* dimensions, the values of the chemical potential is given as:(17)$$\begin{array}{ccc} & & \tilde{\mu }\approx \frac{\alpha d}{8}-T\ln\left(1+\frac{1}{N}\right),\\ & & {\tilde{\mu }}^{\prime }\approx \frac{\alpha d}{2}-T\ln\left(1+\frac{2^{d}}{N}\right). \end{array}$$
Further note that, in the limit N→∞, the values of $\tilde{\mu }$ limit the respective ground state energies from below. Unlike fermions, the ground state occupation of bosons contributes to the chemical potential predominantly in the low-temperature limit. Since inserting the barrier(s) significantly changes the ground state, the chemical potentials with, and without, the barrier(s) are different.

The chemical potential for a Bose gas is slightly less than the ground state energy at a very low temperature, and the difference between them is of the order of inverse volume of the system [53,77]. It goes to zero at a finite temperature only if the volume of the system becomes large enough that the energy level spacing becomes continuous, i.e., α→0, forming free particles and, hence, Bose–Einstein condensate (BEC) is formed at a finite temperature for d≥3. In the limit α→0, as explained before, the quantum advantage is lost, implying that quantum thermal machines based on quantized energy levels and BEC cannot be achieved in the same limit. A few proposals for quantum heat engines or refrigerators that use the BEC phase of the working medium or the bath have been made [78,79], but their thermodynamic advantage is not related to quantized energy levels as they are in this case.

## 3. Results

It is often useful to express the work done in terms of relative partition functions, i.e., the ratios of the partition functions for the adjacent stages, i.e., $\zeta \left(T_{h}\right)=Z_{B}/Z_{A}$ and $\zeta \left(T_{c}\right)=Z_{C}/Z_{D}$. Then, the work done, given by Equation (Equation 5), can be rewritten as:(18)$$\begin{array}{c} W=-(T_{h}\ln\zeta \left(T_{h}\right)-T_{c}\ln\zeta \left(T_{c}\right)). \end{array}$$
The relative partition function is the ratio between the partition functions of the system with, and without, the barrier(s) at the same temperature and, thus, implies the work done due to inserting the barrier(s) to deform the wavefunctions to create a new set of degenerate energy levels [13]. This insertion of the barrier(s) accounts for an extra amount of work −Tlnζ(T). Given a pair of bath temperatures $\left(T_{h},T_{c}\right)$, one can compute the total work done in a cycle using Equation (Equation 18) in terms of the relative partition functions. For ζ(T)=1, the insertion of the barrier(s) did not contribute to any extra work, and, therefore, no advantage was obtained as we expected in the classical limit. In what follows, before we discuss the quantum regime, we briefly review the well-known results of the classical limit for the sake of completeness.

In the classical regime, i.e., T≫α, the thermodynamics of the particles are governed by Maxwell–Boltzmann statistics. The single particle thermal partition function in this regime, in *d* dimensions, is given by:(19)$$\begin{array}{c} Z_{T}=\sum_{n_{x_{1}}=1}^{\infty }\ldots \sum_{n_{x_{d}}=1}^{\infty }e^{-\tilde{E}\left(n_{x_{1}},\ldots ,n_{x_{d}}\right)/T}. \end{array}$$
In terms of the eigenenergies given in Equation (Equation 11), one readily finds the expression for the relative partition function:(20)$$\begin{array}{ccc} \zeta (T)= & & \frac{\sum_{n_{x_{1}}=1}^{\infty }\ldots \sum_{n_{x_{d}}=1}^{\infty }e^{-\alpha (n_{x_{1}}^{2}+\ldots +n_{x_{d}}^{2})/8T}}{2^{d}\sum_{n_{x_{1}}=1}^{\infty }\ldots \sum_{n_{x_{d}}=1}^{\infty }e^{-\alpha (n_{x_{1}}^{2}+\ldots +n_{x_{d}}^{2})/2T}}\approx \frac{{\left(\frac{1}{2}\sqrt{\frac{8\pi T}{\alpha }}\right)}^{d}}{2^{d}\times {\left(\frac{1}{2}\sqrt{\frac{2\pi T}{\alpha }}\right)}^{d}}=1. \end{array}$$

This implies that introducing barrier(s) at x=0 does not cost any extra work in the classical limit. In view of Equations (Equation 18) and (Equation 20), one concludes that the work done (extracted) by (from) the system is identically zero in the classical limit. Therefore, one can easily see that the counterpart of such a quantum cycle operating between two thermal baths, is an incompressible classical engine/refrigerator with zero efficiency [12,56,57]. This paradigm breaks down in the quantum domain, where the discreteness in the energy levels and the inhomogeneous shift of the population distribution can lead to efficient quantum thermal machines with no classical analog [12,13].

First, let us consider the case with particles in a one-dimensional potential well. The relative partition functions from Equation (Equation 12) for fermions and bosons are found to be:(21)$$\begin{array}{c} \zeta^{+}\left(T\right)=\prod_{n=1}^{\infty }\frac{\left(1+e^{-(\alpha {\left(2n\right)}^{2}/8-{\tilde{\mu }}^{\prime })/T}\right)}{\left(1+e^{-(\alpha {\left(2n-1\right)}^{2}/8-\tilde{\mu })/T}\right)}, \end{array}$$
and
(22)$$\begin{array}{c} \zeta^{-}\left(T\right)=\prod_{n=1}^{\infty }\frac{\left(1-e^{-(\alpha {\left(2n-1\right)}^{2}/8-\tilde{\mu })/T}\right)}{\left(1-e^{-(\alpha {\left(2n\right)}^{2}/8-{\tilde{\mu }}^{\prime })/T}\right)}. \end{array}$$
The expressions for the internal energies at *B* and *D* for fermions and bosons are, respectively, given by:(23)$$\begin{array}{ccc} & & U_{B}^{\pm }=\sum_{n=1}^{\infty }2\frac{\alpha n^{2}/2-{\tilde{\mu }}^{\prime }}{e^{\left(\alpha n^{2}/2-{\tilde{\mu }}^{\prime }\right)/T_{h}}\pm 1};\\ & & U_{D}^{\pm }=\sum_{n=1}^{\infty }\frac{\alpha n^{2}/8-\tilde{\mu }}{e^{\left(\alpha n^{2}/8-\tilde{\mu }\right)/T_{c}}\pm 1}. \end{array}$$

We now generalize the result given in the previous section for a quantum gas in a *d*-dimensional box, with a barrier in each dimension. As discussed in Section 2.2, this renders an energy level structure with a degeneracy of $2^{d}$. The expressions for the relative partition functions of the particles are modified according to Equation (Equation 11) as:(24)$$\begin{array}{c} \zeta^{+}\left(T\right)=\prod_{n_{x_{i}}=1}^{\infty }\prod_{j_{x_{i}}=0}^{1}\frac{\left(1+e^{-(\alpha \sum_{i}\left({\left(2n_{x_{i}}\right)}^{2}/8-{\tilde{\mu }}^{\prime }\right)/T}\right)}{\left(1+e^{-(\alpha \sum_{i}\left({\left(2n_{x_{i}}-j_{x_{i}}\right)}^{2}/8-\tilde{\mu }\right)/T}\right)}, \end{array}$$
for fermions and
(25)$$\begin{array}{c} \zeta^{-}\left(T\right)=\prod_{n_{x_{i}}=1}^{\infty }\prod_{j_{x_{i}}=0}^{1}\frac{\left(1-e^{-(\alpha \sum_{i}{\left(2n_{x_{i}}-j_{x_{i}}\right)}^{2}/8-\tilde{\mu })/T}\right)}{\left(1-e^{-(\alpha \sum_{i}{\left(2n_{x_{i}}\right)}^{2}/8-{\tilde{\mu }}^{\prime })/T}\right)}, \end{array}$$
for bosons, respectively.

The extension for the *d*-dimensional internal energies at *B* and *D*, for a system of fermions and bosons, are, respectively, given by:(26)$$\begin{array}{ccc} & & U_{B}^{\pm }=\sum_{n_{x_{i}}=1}^{\infty }2^{d}\frac{\alpha \left(\sum_{i}{\left(2n_{x_{i}}\right)}^{2}\right)/8-{\tilde{\mu }}^{\prime }}{e^{\left(\alpha \left(\sum_{i}{\left(2n_{x_{i}}\right)}^{2}\right)/8-{\tilde{\mu }}^{\prime }\right)/T_{h}}\pm 1},\\ & & U_{D}^{\pm }=\sum_{n_{x_{i}}=1}^{\infty }\frac{\alpha \left(\sum_{i}n_{x_{i}}^{2}\right)/8-\tilde{\mu }}{e^{\left(\alpha \left(\sum_{i}n_{x_{i}}^{2}\right)/8-\tilde{\mu }\right)/T_{c}}\pm 1}. \end{array}$$

To see the contrasting role of the chemical potential on the relative partition functions for fermions and bosons, the temperature dependence of $\zeta^{\pm }\left(T\right)$ was plotted in Figure 2 following Equations (Equation 24) and (Equation 25) for d=1,2 and 3. It is evident from Figure 2a that the dimension of the system of fermions decided the qualitative behavior of $\zeta^{+}\left(T\right)$. The value of $\zeta^{+}\left(T\right)$ was zero at *T* and tended to zero irrespective of *d* but showed different behavior depending on the *d*, owing to the particle number dependence of $\tilde{\mu }$ ($∼{\tilde{E}}_{F}$). In T/α≲1 regime, the value of $\zeta^{+}\left(T\right)$ did not change appreciably from zero for d=1 ($\zeta^{+}\left(T\right)∼10^{-10}$ at T=1), but increased for d=2 and d=3. However, the value of $\zeta^{+}\left(T\right)$ for d=3 was much smaller compared to d=2. As we explained earlier, the relative partition function was related to the amount of work −Tlnζ(T) to change the wavefunctions while inserting the barrier. This work done was always non-zero, irrespective of *d*. On the other hand, for bosons, the qualitative behavior of $\zeta^{-}\left(T\right)$ was similar, irrespective of the dimension, as seen in Figure 2b. The value of $\zeta^{-}\left(T\right)$ was the maximum at T=0 and then decayed to zero quickly, irrespective of the dimension, as the lowest energy level was filled with all the particles, but the value at T=0 increased with *d*, mainly owing to the values of dimension-dependent chemical potentials, given in Equation (Equation 17):

It is further evident from Figure 2 that low-temperature behavior of fermions and bosons would be significantly contrasting, as the relative partition functions for fermions and bosons tended to zero, and finite nonzero values, respectively, as T→0 and their slopes were positive and negative, respectively. It was also expected that the behavior of both fermions and bosons would strongly depend on the dimension of the system. These corollaries solely followed particle statistics and were independent of the specific form of the cycle. Therefore, it captured the generic nature of the low-temperature behavior of any quantum thermal machines with quantum gases. We show in the next subsection, in detail, that our results with the Stirling cycle were consistent with these observations.

Given the two baths with two different temperatures and a system with arbitrary dimensions and a number of particles, it was not straightforward to simplify the above expressions to predict the behavior of the cycle. One could easily plot the expressions to observe the characteristics of the system for a given pair of baths set at two arbitrary temperatures. However, in the next subsection, we elaborate that it was actually possible to look at the extremely low temperature of the system in N→∞ limit analytically, and show the contrasting behavior of the system with fermions and bosons in different dimensions.

### 3.1. Analytical Approach for Extremely Low Temperature & N→∞ Limit of the Cycle

We were specifically interested in the behavior of the Stirling cycle in the quantum regime, i.e., in the extremely low temperatures, given by $T_{h}\to 0,T_{c}\to 0$ with $\Delta T=(T_{h}-T_{c})>0$. In this limit the expression for work (Equation (Equation 18)) reduces to:(27)$$W=\lim_{T\to 0,\Delta T\to 0}-(\left(T+\Delta T\right)\ln\zeta \left(\left(T+\Delta T\right)\right)+T\ln\zeta \left(T\right).$$
Now, let us define the following function of temperature:(28)ω(T)=Tlnζ(T).
The slope of this function solely determines whether, at low temperatures, work can be extracted from the system or not. Work can be extracted from the system only if the slope of the function, i.e., $\lim_{T\to 0}\frac{d}{dT}\omega \left(T\right)$ as T→0 is positive and work is done on the system if the same is negative. Now, in the thermodynamic limit, i.e., N→∞, one can analytically evaluate the sign of the above quantity and, thereby, decide the nature of the system. With reference to Figure 2, one can find that the slopes of the relative function for fermions and bosons were opposite in the low-temperature regime and predicted their distinctive behaviors.

#### 3.1.1. Fermions

Let us first consider the case of non-interacting fermions in a one-dimensional potential well. The function ω(T) takes the form of (cf. Equation (Equation 21)):(29)$$\begin{array}{c} \omega \left(T\right)=T\ln\prod_{n=1}^{\infty }\frac{\left(1+e^{-\alpha ({\left(2n\right)}^{2}-N^{2}/)/8T}\right)}{\left(1+e^{-\alpha ({\left(2n-1\right)}^{2}-N^{2})/8T}\right)}. \end{array}$$

In the aforementioned limiting cases, both the quantities $\lim_{T\to 0}\lim_{N\to \infty }e^{\frac{\alpha }{8T}\left(4n^{2}-N^{2}\right)}$ and $\lim_{T\to 0}\lim_{N\to \infty }e^{\frac{\alpha }{8T}\left({\left(2n-1\right)}^{2}-N^{2}\right)}$ are very small positive numbers and
$$e^{\frac{\alpha }{8T}\left(4n^{2}-N^{2}\right)}>e^{\frac{\alpha }{8T}\left({\left(2n-1\right)}^{2}-N^{2}\right)}\;\;\;\forall \;n.$$
Hence, expanding $\frac{d}{dT}\omega \left(T\right)$ series of *n* we obtain:(30)$$\begin{array}{c} \frac{d}{dT}\omega \left(T\right)=\frac{\alpha }{8T}\sum_{n=1}^{\infty }\left(N^{2}-4n^{2}+1\right)\left[e^{\frac{\alpha }{8T}\left(4n^{2}-N^{2}\right)}\right.\\ \left.-e^{\frac{\alpha }{8T}\left({\left(2n-1\right)}^{2}-N^{2}\right)}\right]+\left(1-4n\right)e^{\frac{\alpha }{8T}\left({\left(2n-1\right)}^{2}-N^{2}\right)}. \end{array}$$
Clearly, for N→∞, the quantity $\frac{d}{dT}\omega \left(T\right)$ is positive. This implies that work can be extracted from a system with a large number of non-interacting fermions in low-temperature limits. Therefore, from Table 1, one concludes that the system behaves exclusively as a heat engine.

The efficiency of the heat engine is then given as:(31)$$\begin{array}{c} \eta_{E}=-\frac{W}{Q_{h}}=\frac{1-\frac{T_{c}\ln\zeta^{+}\left(T_{c}\right)}{T_{h}\ln\zeta^{+}\left(T_{h}\right)}}{1+\frac{U_{B}^{+}-U_{D}^{+}}{T_{h}\ln\zeta^{+}\left(T_{h}\right)}}. \end{array}$$
In the aforementioned limit, the quantities $T_{h}\ln\zeta^{+}\left(T_{h}\right)\to -\infty$ and $\frac{T_{c}\ln\zeta^{+}\left(T_{c}\right)}{T_{h}\ln\zeta^{+}\left(T_{h}\right)}\to \frac{T_{c}}{T_{h}}$. Therefore, efficiency $\eta_{E}\to 1-\frac{T_{c}}{T_{h}}$, i.e., the Carnot limit. The above equation connotes that a large number of non-interacting fermions entrapped in a one-dimensional potential well behave like a heat engine, and its efficiency tends to the Carnot limit as the bath temperatures approach absolute zero. Here, we restate that it is purely a quantum effect with no classical correspondence, yet the machine still works with the highest possible efficiency and abides by the second law of thermodynamics. This is the first important result of our analysis, exhibiting the true quantum signature at a macroscopic scale.

However, for fermions at higher dimensions, the behavior of the system is entirely different, because of the different number dependence of the chemical potential. The function ω(T), in this case, is given by (cf. Equation (Equation 24)):(32)$$\begin{array}{c} \omega \left(T\right)=T\ln\prod_{n_{x_{i}}=1}^{\infty }\prod_{j_{x_{i}}=0}^{1}\frac{\left(1+e^{-(\alpha \sum_{i}{\left(2n_{x_{i}}\right)}^{2}/8-\tilde{\mu })/T}\right)}{\left(1+e^{-(\alpha \sum_{i}{\left(2n_{x_{i}}-j_{x_{i}}\right)}^{2}/8-\tilde{\mu })/T}\right)}. \end{array}$$
We know that in the low-temperature limit, the chemical potential $\tilde{\mu }$ can equal the Fermi energy ${\tilde{E}}_{F}$. It can be seen from Equation (Equation 13) that the chemical potential in higher dimensions varies in a sub-quadratic fashion with *N* as $\tilde{\mu }\propto N^{\gamma }$, γ<2, whereas the energy dispersion is still quadratic, as given in Equation (Equation 11). As a result, in the macroscopic thermodynamic limit N→∞, the energy always outgrows the chemical potential. Now, in the limit N→∞ and T→0, the expression $\frac{1}{T}\left(\alpha \sum_{i}{\left(2n_{x_{i}}\right)}^{2}/8-\tilde{\mu }\right)$ tends to *∞* or −∞ depending on the values of $\tilde{\mu }$. If $\frac{1}{T}\left(\alpha \sum_{i}{\left(2n_{x_{i}}\right)}^{2}/8-\tilde{\mu }\right)$ tends to *∞*, then $e^{-\frac{1}{T}\left(\alpha \sum_{i}{\left(2n_{x_{i}}\right)}^{2}/8-\tilde{\mu }\right)}\to 0$. On the other hand, if $\frac{1}{T}\left(\alpha \sum_{i}{\left(2n_{x_{i}}\right)}^{2}/8-\tilde{\mu }\right)$ tends to −∞,, then $e^{\frac{1}{T}\left(\alpha \sum_{i}{\left(2n_{x_{i}}\right)}^{2}/8-\tilde{\mu }\right)}\to 0$. So, we split up the sum over $n_{x_{i}},j_{x_{i}}$ in two parts and denote them by $\sum_{1}$ and $\sum_{2}$ for these two cases, respectively. Hence:(33)$$\begin{array}{ccc} & & \frac{d}{dT}\omega \left(T\right)=\sum_{1}\left[e^{-F_{1}\left(n_{x_{i}},j_{x_{i}}\right)}-e^{-F_{2}\left(n_{x_{i}},j_{x_{i}}\right)}\right]\\ & & +\sum_{1}\left[F_{1}\left(n_{x_{i}},j_{x_{i}}\right)e^{-F_{1}\left(n_{x_{i}},j_{x_{i}}\right)}-F_{2}\left(n_{x_{i}},j_{x_{i}}\right)e^{-F_{2}\left(n_{x_{i}},j_{x_{i}}\right)}\right]\\ & & -\sum_{2}\left[F_{1}\left(n_{x_{i}},j_{x_{i}}\right)e^{F_{1}\left(n_{x_{i}},j_{x_{i}}\right)}-F_{2}\left(n_{x_{i}},j_{x_{i}}\right)e^{F_{2}\left(n_{x_{i}},j_{x_{i}}\right)}\right], \end{array}$$
where, $F_{1}\left(n_{x_{i}},j_{x_{i}}\right)=\frac{1}{T}\left(\frac{\alpha }{8}\sum_{i}{\left(2n_{x_{i}}\right)}^{2}-\tilde{\mu }\right)$ and $F_{2}\left(n_{x_{i}},j_{x_{i}}\right)=\frac{1}{T}\left(\frac{\alpha }{8}\sum_{i}{\left(2n_{x_{i}}-j_{x_{i}}\right)}^{2}-\tilde{\mu }\right)$. Using the following inequalities for each set of integers $\left(n_{x_{1}},n_{x_{2}},\ldots \right)$ and $j_{x_{i}}=0,1$,
(34)$$\begin{array}{ccc} & & e^{\frac{1}{T}\left(\frac{\alpha }{8}\sum_{i}{\left(2n_{x_{i}}\right)}^{2}-\tilde{\mu }\right)}\geq e^{\frac{1}{T}\left(\frac{\alpha }{8}\sum_{i}{\left(2n_{x_{i}}-j_{x_{i}}\right)}^{2}-\tilde{\mu }\right)},\\ & & e^{-\frac{1}{T}\left(\frac{\alpha }{8}\sum_{i}{\left(2n_{x_{i}}-j_{x_{i}}\right)}^{2}-\tilde{\mu }\right)}\geq e^{-\frac{1}{T}\left(\frac{\alpha }{8}\sum_{i}{\left(2n_{x_{i}}\right)}^{2}-\tilde{\mu }\right)}, \end{array}$$
one finds that the first and the third terms are negative and positive, respectively, and the second term can be both positive and negative. Therefore, $\frac{d}{dT}\omega \left(T\right)$ can be both positive and negative, depending on the actual functional forms of the chemical potential and dimension. Therefore, the non-interacting fermions at low temperatures in more than one dimension can behave as a heat engine or a refrigerator, accelerator, and heater, depending on the sign of $Q_{h}$ and $Q_{c}$. The interplay between the number dependence on the chemical potential and the energy levels in d>2 dictates the behavior of fermions, which we explicitly show in numerical results later. This was the second important observation of our analysis.

#### 3.1.2. Bosons

Now, we consider the case of non-interacting bosons in a one-dimensional box. The function ω(T) takes the form of (Equation (Equation 22)):(35)$$\begin{array}{c} \omega \left(T\right)=T\ln\prod_{n=1}^{\infty }\frac{\left(1-e^{-(\alpha {\left(2n-1\right)}^{2}/8-\tilde{\mu })/T}\right)}{\left(1-e^{-(\alpha {\left(2n\right)}^{2}/8-{\tilde{\mu }}^{\prime })/T}\right)}, \end{array}$$

Expanding $\frac{d}{dT}\omega \left(T\right)$ in the T→0 limit we obtain:(36)$$\begin{array}{ccc} & & \frac{d}{dT}\omega \left(T\right)=\sum_{n=1}^{\infty }\ln\frac{\left(1-e^{-(\alpha {\left(2n-1\right)}^{2}/8-\tilde{\mu })/T}\right)}{\left(1-e^{-(\alpha {\left(2n\right)}^{2}/8-{\tilde{\mu }}^{\prime })/T}\right)}\\ & & +\frac{1}{T}\sum_{n=1}^{\infty }(\left(\frac{\alpha }{8}{\left(2n\right)}^{2}-{\tilde{\mu }}^{\prime }\right)e^{-(\alpha {\left(2n\right)}^{2}/8-{\tilde{\mu }}^{\prime })/T}\\ & & -\left(\frac{\alpha }{8}{\left(2n-1\right)}^{2}-\tilde{\mu }\right)e^{-(\alpha {\left(2n-1\right)}^{2}/8-\tilde{\mu })/T}). \end{array}$$

In the N→∞ limit, using the inequality
(37)$$\begin{array}{c} \frac{\alpha }{8}{\left(2n\right)}^{2}-{\tilde{\mu }}^{\prime }>\frac{\alpha }{8}{\left(2n-1\right)}^{2}-\tilde{\mu }\;\;\forall n\geq 1 \end{array}$$
with μ=α/8 and $\mu^{\prime }=\alpha /2$ (cf. Equation (Equation 17)), one readily shows the quantity dω(T)/dT to be negative and, therefore, implies that a system with a large number of non-interacting bosons in d=1, behaves as a refrigerator or an accelerator or a heater, depending on the signs of $Q_{h}$ and $Q_{c}$.

However, for bosons at higher dimensions, the function ω(T), in this case, is given by (cf. Equation (Equation 25)):(38)$$\begin{array}{c} \omega \left(T\right)=T\ln\prod_{n_{x_{i}}=1}^{\infty }\prod_{j_{x_{i}}=0}^{1}\frac{\left(1-e^{-(\alpha \sum_{i}{\left(2n_{x_{i}}-j_{x_{i}}\right)}^{2}/8-\tilde{\mu })/T}\right)}{\left(1-e^{-(\alpha \sum_{i}{\left(2n_{x_{i}}\right)}^{2}/8-{\tilde{\mu }}^{\prime })/T}\right)}. \end{array}$$

In the limit T→0, unlike fermions, only the ground state is occupied by the bosons, and, consequently, the occupation at the excited states is negligible. In the N→∞ limit, we use the following inequality for the set of integers $n_{x_{i}}=1$ and $j_{x_{i}}=0,1$
(39)$$\begin{array}{c} \frac{\alpha }{8}\left(\sum_{i}{\left(2n_{x_{i}}\right)}^{2}\right)-{\tilde{\mu }}^{\prime }\leq \frac{\alpha }{8}\left(\sum_{i}{\left(2n_{x_{i}}-j_{x_{i}}\right)}^{2}\right)-\tilde{\mu },\;\;\forall d\geq 2, \end{array}$$
with $\tilde{\mu }=\alpha d/8$ and ${\tilde{\mu }}^{\prime }=\alpha d/2$, one finds that dω/dT is positive (note that the inequalities in Equations (Equation 37) and (Equation 39) have reversed signs). This implies that the system with non-interacting bosons in d≥2 behaves as a heat engine in the low-temperature limit. The efficiency of the heat engine is then given as:(40)$$\begin{array}{c} \eta_{E}=-\frac{W}{Q_{h}}=\frac{1-\frac{T_{c}\ln\zeta^{-}\left(T_{c}\right)}{T_{h}\ln\zeta^{-}\left(T_{h}\right)}}{1+\frac{U_{B}^{-}-U_{D}^{-}}{T_{h}\ln\zeta^{-}\left(T_{h}\right)}}. \end{array}$$
In the limit $T_{h},T_{c}\to 0$, both the quantities $\zeta^{-}\left(T_{h}\right)$ and $\zeta^{-}\left(T_{c}\right)$ are nonzero, $\ln\zeta^{-}\left(T_{c}\right)>\ln\zeta^{-}\left(T_{h}\right)$ (Figure 2b) and $U_{B}^{-}>U_{D}^{-}$ ensure an efficiency less than the Carnot bound, unlike the fermions in d=1.

In Figure 3, we present different modes of operations for fermions in one, two, and three dimensions for N=20 and 40 from numerical calculations. Fermions in a one-dimensional box behaved strictly as a heat engine in the $T_{h}-T_{c}$ parameter space (Figure 3a,b), as predicted from the analytical approach. The efficiency was almost constant and equal to the Carnot boundary in the low-temperature regime $T_{h}≲\alpha$. On the other hand, in d=2 and d=3, we predicted, from Equation (Equation 33), that all the four modes could coexist in the $T_{h}-T_{c}$ plane in the low-temperature limit. In d=2, from numerical calculations, we also found that the three modes (refrigerator, accelerator, and heater) with W>0 for N=20, and all the four modes (engine, refrigerator, accelerator, and heater) with N=40, co-existed in the $T_{h}-T_{c}$ plane (Figure 3c,d). In d=3, we found that all the four modes (refrigerator, engine, heater, and accelerator) for N=20 and the three modes (refrigerator, heater, and accelerator) with W>0 for N=40 co-existed (Figure 3e,f). In all the cases above, when $T_{c}≲T_{h}$, the system worked as a refrigerator, but as one increased $T_{h}$, while keeping $T_{c}$ fixed, the region of the accelerator was reached. A narrow region of the heater demarcated the boundary between the regions of the refrigerator and accelerator, where $Q_{h}$ flipped its sign. The numerical results concluded that the low-temperature behavior of fermions for d>1 depended both on the dimensions and the number of particles in the system, owing to the interplay between the number dependence of the chemical potential and energy levels, as explained through our analytical approach.

In contrast to fermions, as predicted from our analytical approach, the system with bosons in d=1 predominantly behaved as an accelerator with small regions of refrigerator and heater (Figure 4a,b). However, it showed qualitatively different behavior in d=2,3 (Figure 4c–f), where all four distinct regions existed, but near $T_{h},T_{c}\to 0$, the system behaved as a heat engine, as predicted from the analysis. We showed the results with two different values of *N*; Figure 4a–e for N=20 and Figure 4b–f for N=40, but no qualitative difference depending on *N* was observed.

The following common remarks can be made from the results shown in Figure 3 and Figure 4: (a) The fermi gas in one dimension was the most desired system to construct a heat engine with Carnot efficiency. However, a boson gas was also a candidate for heat engines, provided the dimension of the system was greater than one; (b) The region of the refrigerator mode was located near the boundary $T_{h}=T_{c}$ and the region of the accelerator on the other boundary $T_{c}∼0$ for bosons and fermions in d=2,3. A small region of the heater demarcated the boundary of transition between the refrigerator and the accelerator. Therefore, the refrigerator was not a suitable mode for quantum gases, in general.

### 3.2. Dependence on Average Particle Number *N*

We have already shown that a Stirling cycle with fermions in one dimension exclusively worked as an engine, whereas with bosons, it both behaved as a refrigerator and a heater. To explore the *N* dependence for fermions, we plotted the heat engine efficiency scaled w.r.t the Carnot efficiency, $\eta_{E}/\eta_{E}^{max}$ in Figure 5a, with the number of particles *N* and $T_{h}$, in an one-dimensional box, keeping the ratio $T_{c}/T_{h}=0.5$ fixed. The efficiency reached the Carnot bound as the bath temperatures tended to zero. It is interesting to see that for a given pair of baths with non-zero temperatures, a system with larger *N* yielded better efficiency. Owing to a prominent dependence on fermions, the engine efficiency could be boosted by adding more fermions to the system.

In Figure 5b, we plotted the engine efficiency for bosons scaled w.r.t the Carnot efficiency, with the number of particles *N*, and $T_{h}$, for d=2,3, keeping the ratio $T_{c}/T_{h}=0.5$ fixed. The scaled efficiency tended to values much less than 1 as the bath temperatures tended to zero, irrespective of *N* and *d*, as predicted from our analysis, and it also went to zero when the system switched to the accelerator mode, as shown already in Figure 4c–f. The maximum coefficient of performance increased slightly with *N*, as seen from Figure 5b, revealing that bosons did not show any prominent dependence on *N*, unlike fermions.

## 4. Conclusions

To conclude, we demonstrated some fundamental features of quantum particles in the context of the quantum Stirling cycle. Given a thermodynamics cycle, one can anticipate that the behavior of the working medium in the quantum regime is fundamentally determined only by the quantum statistics of the particles. We analyzed the role of quantum statistics and system dimensions in determining the collective thermodynamic behavior of particles. It is worth mentioning that, the efficiency considered here exhibited a pure quantum signature, with no classical analog, and manifested only in the quantum regime. Though our analysis focused on a specific working medium, viz., quantum gas trapped in an infinite potential well or a square well, one could trivially generalize our analysis for any system. A similar analysis with other trapping potentials would connote a qualitatively similar result, as a consequence of the statistical nature of identical quantum particles.

We found that bosons behaved completely opposite to fermions as a manifestation of the fundamental difference between the particle statistics in the quantum domain. Therefore, fermions and bosons are useful in a different way from the viewpoint of constructing a desirable thermodynamic machine. A Stirling cycle with fermions confined in a one-dimensional potential well, when connected to a pair of low-temperature baths, behaves exclusively like a heat engine (W<0), while a Stirling cycle with bosons behaves like an accelerator, refrigerator and a heater (W>0), depending on the bath temperatures. On the other hand, the behaviour of bosons and fermions is qualitatively similar in two and three-dimensional systems. In both cases, unlike in one dimension, a mixture of four different modes can be observed, depending on the bath temperatures and the number of particles.

We also showed that the particle number or the dimension does not appreciably affect the performance of bosons. Unlike bosons, the number dependence of the chemical potential for fermions decides the behavior of the system. As the dimension of the system decides the number dependence of the chemical potential, it determines the overall thermodynamic behavior of the system. The behavior of fermions and bosons in a one-dimensional well was completely different compared to that of two or higher-dimensional boxes. We found that increasing the number of particles in a system could boost its performance in spite of the engine efficiency/coefficient of performance being bounded by the Carnot bound.

## Figures and Tables

**Figure 1 entropy-25-00372-f001:**
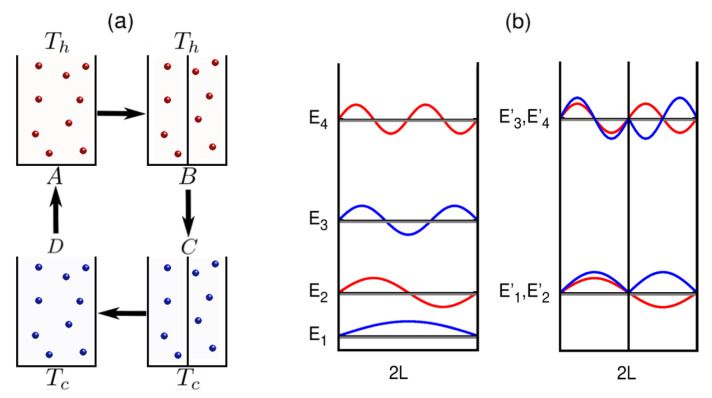
(**a**) A schematic representation of a Stirling engine with *N* massive quantum particles confined in an infinite potential box. (**b**) Single particle energy levels and wavefunctions of a particle in an infinite potential well with, and without, the barrier.

**Figure 2 entropy-25-00372-f002:**
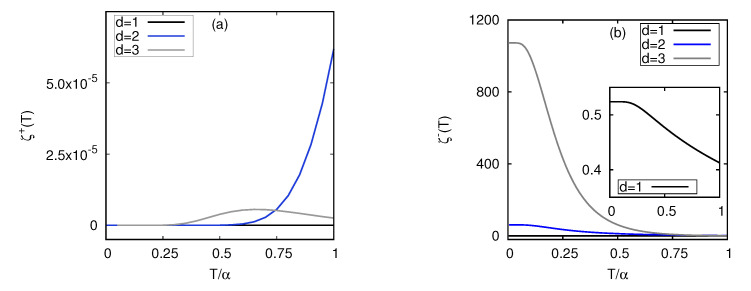
Relative partition function $\zeta^{\pm }\left(T\right)$ (cf. Equations (Equation 24) and (Equation 25)) as a function of *T* for (**a**) fermions and (**b**) bosons with N=20.

**Figure 3 entropy-25-00372-f003:**
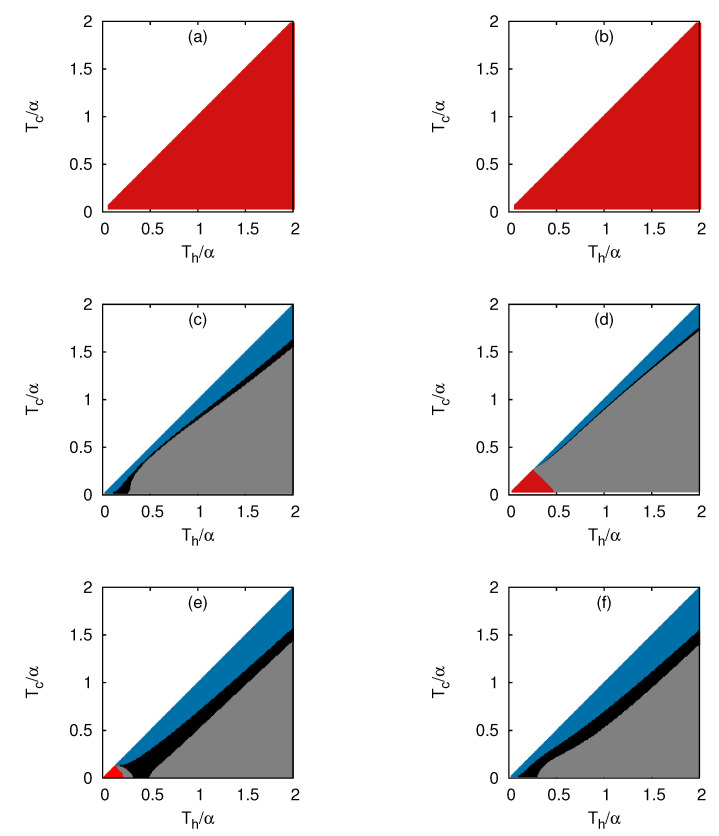
Different modes of operation of a Stirling engine with a gas of fermions. The modes of operation as functions of $T_{h}$ and $T_{c}$ for: (**a**) d=1, N=20; (**b**) d=1, N=40; (**c**) d=2, N=20; (**d**) d=2, N=40; (**e**) d=3, N=20; (**f**) d=3, N=40. The regimes corresponding to Engine, Refrigerator, Accelerator, and Heater are marked with red, blue, grey, and black, respectively.

**Figure 4 entropy-25-00372-f004:**
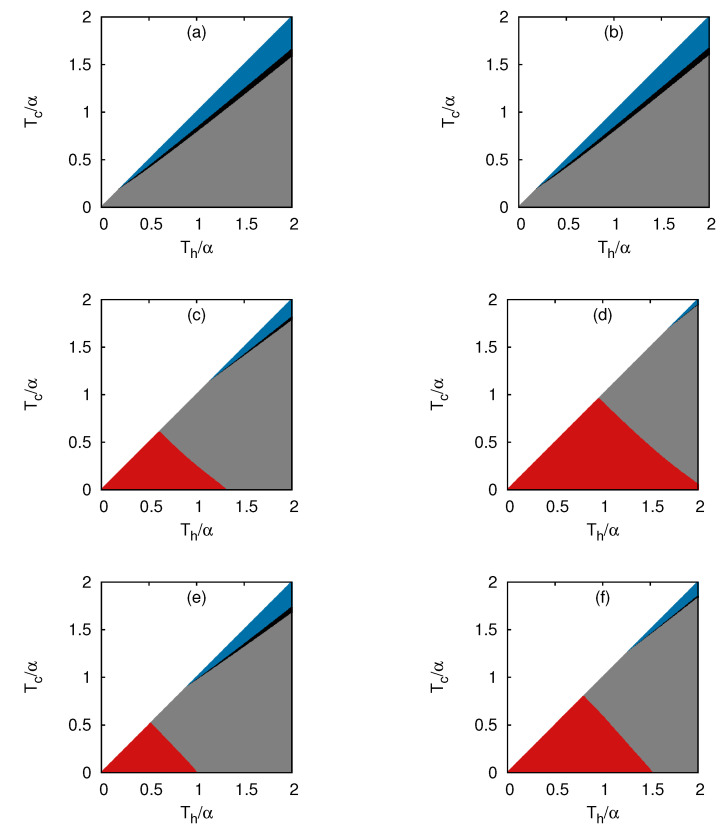
Different modes of operation of a Stirling engine with a gas of bosons. The modes of operation as functions of $T_{h}$ and $T_{c}$ for: (**a**) d=1, N=20; (**b**) d=1, N=40; (**c**) d=2, N=20; (**d**) d=2, N=40; (**e**) d=3, N=20; (**f**) d=3, N=40. The regimes corresponding to Engine, Refrigerator, Accelerator, and Heater are marked with red, blue, grey, and black, respectively.

**Figure 5 entropy-25-00372-f005:**
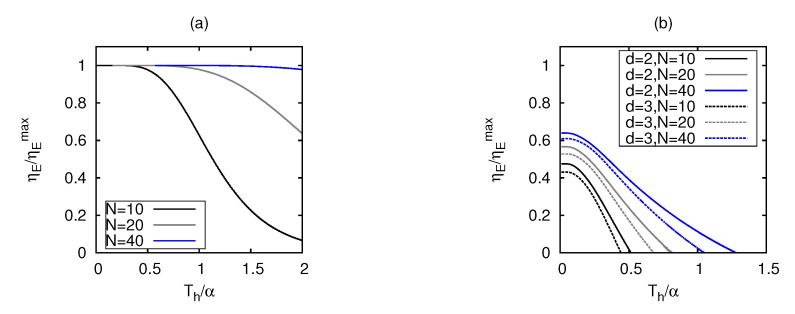
Dependence on the particle number *N* on engine efficiency. Engine efficiency scaled w.r.t. Carnot efficiency $\eta_{E}/\eta_{E}^{max}$ for a fixed $T_{c}/T_{h}=0.50$ for (**a**) fermions in d=1 and (**b**) bosons in d=2,3.

**Table 1 entropy-25-00372-t001:** Different modes of operation of a quantum thermodynamic cycle.

Modes of Operation	$Q_{h}$	$Q_{c}$	*W*
Engine	>0	<0	<0
Refrigerator	<0	>0	>0
Accelerator	>0	<0	>0
Heater	<0	<0	>0

## Data Availability

All data generated or analyzed during this study are available from the authors on reasonable request.

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
