# Peer review of "Quantum Advantage of Thermal Machines with Bose and Fermi Gases"

_entropy, 2023, doi:10.3390/e25020372_

Round 1

Reviewer 1 Report

In this manuscript, the authors consider Bose and Fermi gases as the working media of a quantum Sterling engine, confined in an infinite-potential box with an infinite-potential barrier being quasi-statically created and removed during the process. They discover a dramatic difference between the bosons and fermions due to their distinctive quantum statistics, providing a very rich phase diagram with its operational characterization of the engine, refrigerator, accelerator, and heater. The manuscript is well-written, the calculations are thorough, and the results look solid. I believe their findings contribute significantly to the advance in the thermodynamics of quantum engines. Therefore, I recommend the publication of this manuscript in Entropy.

Just one error to be fixed in the manuscript: there is a missing reference in Line 183.

Author Response

We thankful to the referee for spending her/his precious time in carefully reviewing our manuscript and suggesting for its publication.

Point 1: Just one error to be fixed in the manuscript: there is a missing reference in Line 183.

Response: We have fixed the hyper-links in the revised version.

Reviewer 2 Report

The authors study the cycles of Stirling quantum thermodynamic machine using as a working medium non-interacting particles, which can be fermions or bosons, trapped in a d-dimensional box , where d=1, 2, 3, corresponding to quantum wires, wells and dots, respectively. In the quantum regime of small box sizes and low enough operating temperatures, the authors show analytically and numerically that the operational mode of the machine (heat engine, refrigerator, accelerator, heater) , depends strongly on the nature of the particles (fermions or bosons) and the box dimension, a purely quantum behavior without classical counterpart.

The article is well written, and the presented results are quite interesting and timely, given that there are several experimental platforms which promise the realization of quantum thermal machines. We thus recommend its publication in Entropy, while we only have some minor observations, which may lead in the improvement of the presentation:

1.       State that in Eq. (12) the dependence on the particle number is hidden in the chemical potential.

2.       In Eq. (20) use larger parentheses in both the numerator and the denominator.

3.       In Eq. (21), in the denominator, it is T, not T’.

4.       Ln. 128, “is only a…”

5.       Lns. 128-130, rephrase the statement.

6.       Ln. 146, “has zero contribution to work”.

7.       Ln. 221, delete double “the”.

8.       Lns. 600-603 correct the references.

9.       The authors may would like to cite in the Introduction the following works on the optimization of the quantum harmonic Otto cycle:

(a)    D. Stefanatos, Optimal efficiency of a noisy quantum heat engine, Phys. Rev. E 90, 012119 (2014).

(b)    D. Stefanatos, Exponential bound in the quest for absolute zero, Phys. Rev. E 96, 042103 (2017).  

Reviewer 3 Report

The overall evaluation on the draft is not good, since it’s difficult to find the creativity and ingenuity in it.

Furthermore, the whole clarity of the draft seems poor to convey the physical meaning. More specifically, more comments follow.

The many opinions have been mentioned without specific references. For example in section 1,

The collective behavior of quantum particles at line 35.

Quantum supremacy arising out of the interplay at line 42.

And the followings are with references, but their meanings should be more specific and clearer:

Creating a new energy level structure with degeneracies at line 50.

Bypassing the prohibitive complexity of tracking the individual particles at line 59.

In certain limiting cases at line 73

One suggestion on the draft is that it might be better to put a short comparison paragraph with other cycles though Quantum Stirling cycle is considered in the draft.

We know that the phase transition in low-temperature at Bose gas, which is known as Bose-Einstein condensation, is not mentioned, where the behavior of chemical potential is very different from usual temperature range.

The first sentence at abstract is about thermodynamics of a single quantum particle is not relevant to the draft (at line 1 and 2). Furthermore, in a single particle we cannot assess the spin-statistics from identical particles. So to get better unison of the paper, the abstract might be rewritten in more organized way.

In conclusion, it is claimed that the main result is that the demonstration of “fundamental” features of quantum particles (at line 426). But, the meaning of these features should be described in more specific way. And, the qualitative similarities of bosons and fermions are also clarified (at line 442).

 The overall evaluation for the draft is that it should be improved.
